# Combined Effects of Particle Size and Dough Improvers on Improving the Quality of Purple-Colored Whole Wheat Bread

**DOI:** 10.3390/foods12132591

**Published:** 2023-07-03

**Authors:** Enkhtungalag Avarzed, Meera Kweon

**Affiliations:** 1Department of Food Science and Nutrition, Pusan National University, Busan 46241, Republic of Korea; tungalagtunga45@gmail.com; 2Kimchi Research Institute, Pusan National University, Busan 46241, Republic of Korea

**Keywords:** bread, dough improver, particle size, purple-colored wheat, whole wheat flour

## Abstract

Consumers’ interest in healthy products is increasing. However, the production of excellent-quality whole wheat bread (WWB) faces challenges due to the reduced gluten functionality and varied particle sizes of whole wheat flour (WWF). This study aimed to explore the enhancement of purple-colored WWB quality by controlling the particle size of WWF and using dough improvers. Six purple-colored WWFs were obtained using an ultra-centrifugal mill with different sieve openings (0.5 and 1.0 mm) and rotor speeds (6000, 10,000, and 14,000 rpm). The average particle diameter (d50) of the smaller particle size group (S) and the larger particle size group (L) based on the sieve opening ranged from 115 to 258 μm and 294 to 492 μm, respectively. Group S demonstrated higher water absorption, damaged starch, and gluten strength compared to group L. Additionally, group S exhibited a greater bread volume and height compared to group L. Among the tested dough improvers (vital wheat gluten, vitamin C, enzymes, and emulsifiers), vital wheat gluten was the most effective in improving the quality of purple-colored WWB. The improvement effect was significantly greater in group S than in group L. These findings suggest that controlling the particle size of purple-colored WWFs and utilizing dough improvers can result in superior-quality WWB.

## 1. Introduction

Consumers’ interest in healthy grain products has increased due to the health benefits associated with various nutrients found in whole wheat or wheat bran [1,2]. The bran layer of wheat is a rich source of essential functional components such as vitamins, minerals, phytochemicals, and dietary fiber. In particular, dietary fiber intake has shown promising preventive effects against obesity, diabetes, cardiovascular disease, and gastrointestinal diseases [3,4]. It also promotes the growth of intestinal bifidobacteria, aids in bowel movement, and enhances immunity [5]. To meet this increasing demand, the food industry has developed whole wheat and wheat bran products. However, the production of high-quality products faces challenges due to the variations in particle size and reduced gluten functionality caused by wheat bran. 

Previous studies have reported both positive and negative effects of the particle size in whole wheat flour (WWF) on the quality of whole wheat bread (WWB), specifically in terms of volume, height, and texture [6]. Some studies have indicated that smaller particle sizes result in a reduced bread volume [7], while others suggest that smaller particle sizes lead to an increased volume [8]. Additionally, there are reports indicating that a medium particle size contributes to increased volume [9]. Bressiani et al. [10] observed the negative effects of WWF with small particle sizes compared to WWF with medium or large particle sizes on the dough and bread quality characteristics of different WWF types. The baking performance of WWF with a medium particle size has been identified as the most favorable, suggesting a nonlinear correlation between particle size and bread-making performance. Protonotariou et al. [11] found no significant effect of a particle size between 18 and 84 μm of WWF on bread volume, while Lin et al. [12] observed that smaller particle sizes increased the bread volume. Conversely, Hemdane et al. [13] reported no clear relationship between the wheat bran particle size and bread volume. Consequently, further research is required to thoroughly investigate bread-making performance by preparing WWF with varying particle sizes using the same wheat kernels and milling equipment.

To address the reduced gluten functionality caused by wheat bran, various dough improvers have been employed to improve the quality of whole wheat bread (WWB) [14,15,16,17]. Vital wheat gluten is commonly used to compensate for the reduced gluten strength by diluting gluten in WWF [18]. Emulsifiers can interact with the hydrophobic surfaces of proteins in the gluten network, leading to improved dough strength and gas retention during fermentation, resulting in a desirable bread texture and softer crumbs [19,20]. Enzymes are well-known dough conditioners that enhance the stability and elasticity of dough while exerting antioxidant effects [18]. Ascorbic acid, a reducing agent with potent antioxidant properties, stabilizes the gluten protein network in bread dough [20].

In Korea, wheat consumption as food amounts to approximately 2 million tons per year. However, 99% of the total wheat used is imported from the USA, Australia, and Canada, as Korean domestic wheat production is limited to only around 1% of the total consumption [21]. To increase the production and consumption of domestically grown wheat, a purple-colored wheat variety called ‘Ariheuk’ has been developed through the National Institute of Crop Science breeding program. Ariheuk wheat contains higher levels of anthocyanin and polyphenolic compounds compared to common wheat [22,23,24], making it a value-added wheat variety in terms of promoting health. With growing consumer awareness regarding the nutritional and health aspects of whole wheat products and bread made with wheat bran, the applications of Ariheuk wheat have been expanded and investigated [25,26,27]. Research has shown that the blending of purple-colored Ariheuk wheat bran, even at a 30% ratio, can improve the quality of bread by controlling the water amount, mixing time, and fermentation time [25]. The particle size reduction of bran also enhanced the noodle-making performance of flour blended with purple-colored Ariheuk wheat bran and improved the antioxidant activity of noodles by better embedding into noodle sheets [26]. Optimization of the formula and processing conditions for bread prepared with a 30% blend of purple-colored Ariheuk wheat bran was achieved using a response surface methodology (RSM). The total phenolic and anthocyanin content, as well as the antioxidant activity (ABTS and DPPH radical scavenging activity), of the bread increased proportionally as the bran blending ratio increased [27]. However, WWF definitions, specifications, and standards have not yet been established in Korea, which hinders the development and consumption of whole wheat products [28], despite whole grains being internationally defined as having the same composition ratio of endosperm, germ, and bran as grains before grinding [29]. To produce uniform WWF, it is necessary to establish a milling method to control the particle size, as research has shown that the particle size of wheat bran significantly affects the processing characteristics of WWF [30].

Therefore, this study aimed to investigate the effect of the particle size in purple-colored WWF, prepared using an ultra-centrifugal mill under various conditions, on bread-making performance to produce more suitable WWB. Additionally, the study assessed the effect of dough improvers on WWB quality using purple-colored WWF.

## 2. Materials and Methods

### 2.1. Materials

Wheat kernels of the Korean domestic, purple-colored wheat, namely Ariheuk, used in this study were supplied by the National Institute of Crop Science. The moisture and ash content of whole wheat kernels was 12.8% and 1.4%, respectively. Sugar, shortened dry yeast (Societe Industrielle Lesaffre, France), skim milk powder, and salt were purchased from a local market for bread baking. Vital wheat gluten (Bob’s Red Mill Natural Foods, Inc., Milwaukie, OR, USA), vitamin C (ascorbic acid, DSM Jianshan Pharmaceutical Co., Ltd., Jingjang, China), two emulsifiers DATEM (Panodan, Danisco, Grindsted, Denmark) and SSL (Almax-6900, Ilishinwells, Seoul, Korea), and two enzymes from Novozymes, amylase (Fungamyl 4000) and xylanase (Pentopan 500), were used. Reagent-grade (SRC) sodium carbonate, lactic acid, and sucrose were purchased and used to analyze the solvent retention capacity (SRC). 

### 2.2. Milling Purple-Colored WWF and Measuring Particle Size Distribution

Purple-colored WWF was prepared using an ultra-centrifugal mill (FM200, Beijing Grinder Instruments, Beijing, China) equipped with a 12-tooth rotor and mounted with a sieve without tempering. The tested sieve openings were 0.5 mm and 1.0 mm, and the rotor speeds were 6000 rpm, 10,000 rpm, and 14,000 rpm. WWF pulverized with a 0.5 mm sieve was referred to as group S, and WWF pulverized with a 1.0 mm sieve was referred to as group L. Depending on the rotor speed, group S comprised low (SL), medium (SM), and high (SH) speeds, and group L comprised low (LL), medium (LM), and high (LH) speeds, at 6000, 10,000, and 14,000 rpm, respectively. A particle size analyzer (LS 13 320, Beckman Coulter, Brea, CA, USA) was used to measure the particle size distribution of purple-colored WWF using the dry method.

### 2.3. Analyzing Quality Characteristics of Purple-Colored WWF

The quality characteristics of purple-colored WWF were measured using SRC in water and sodium carbonate solutions according to AACC Method 56–11.02 [31], as lactic acid and sucrose solutions have been reported to generate errors due to the bran contained in WWF [32].

The sodium dodecyl sulfate (SDS) sedimentation volume was measured to determine the relative gluten strength in the purple-colored WWF, according to AACC Method 56–70.01 [24]. Purple-colored WWF (5 g) was placed in a 100 mL graduated cylinder, and 50 mL of distilled water was added. The cylinder was closed with a stopper and shaken vertically and horizontally for flour dispersion and hydration. After adding 50 mL of SDS–lactic acid solution (3%), the cylinder was shaken again vertically and horizontally and allowed to stand upright. The volume (mL) of flour precipitated after 20, 40, and 60 min was recorded.

### 2.4. Measuring the Dough-Mixing Property of Purple-Colored WWF Using a Mixograph

The dough-mixing property of purple-colored WWF was measured using a mixograph (10 g, National Manufacturing Co., Lincoln, NE, USA) according to AACC Method 54–40.02 [31]. WWF (10 g) was placed in a mixograph bowl, and 7.0 g and 7.5 g of distilled water was added as the optimal water absorption determined in a preliminary experiment, and the mixture was kneaded for 10 min. The experiment was conducted at a laboratory temperature of 24 °C to minimize variation. Mixograms were obtained and analyzed for dough properties. 

### 2.5. Preparation of WWB with Purple-Colored WWF

WWB made with purple WWF was prepared by slightly modifying AACC Method 10–10.03 [31] (see Table 1 for the formulation). 

After placing flour, salt, shortening, and skim milk powder into a pin mixer bowl (100 g, National Manufacturing, Inc., Lincoln, NE, USA), a pre-mixed yeast and sucrose solution with an amount of water calculated based on the SRC value was added. The mixture was kneaded for 3.5 min. The dough was passed through a dough sheeter (YT-160, Shanghai Huayuan Food Machinery Co., Ltd., Shanghai, China), adjusted to a thickness of 0.47 cm, and then folded and rolled into a baking pan. The pan containing the dough was placed into a fermenter (Phantom M301 Combi, Samjung, Gyeonggi, Korea), adjusted to a temperature of 35 °C and humidity of 85%, and fermented for 50 min. The fermented dough was baked in an oven (Phantom M301 Combi, Samjung) preheated to 215 °C for 18 min. Dough height was measured after fermentation. After baking, the bread was cooled at room temperature for 20 min, removed from the baking pan, cooled for another 30 min, and used to analyze quality characteristics such as bread firmness, volume, height, and color.

### 2.6. Analyzing the Quality Characteristics of Purple-Colored WWB and Changes during Storage

The dough height after fermentation and bread height were measured three times using a caliper (HDS-20C, Mitutoyo, Kanagawa, Japan), and the average value was calculated.

The WWB volume was measured using the rapeseed replacement method (AACC Method 10–05.01) [31], with minor modifications. The bread was placed in a container with a volume of 1.0 L filled with millet. The weight of the millet was measured, and the bread volume was calculated. Measurements were repeated twice for each sample. 

The WWB crumb color was measured using a colorimeter (CR-20, Minolta Co., Ltd., Tokyo, Japan) for *L** (lightness), *a** (redness), and *b** (yellowness) values. The moisture content of the WWB was measured using AACC method 44–15.02 [31]. Briefly, breadcrumbs (3 g) were weighed and placed in a container

Changes in firmness during WWB storage were measured using a Texture Analyzer (Brookfield CT3, Middleboro, MA, USA) according to AACC Method 74–09.01 [31]. Three slices were cut from the center of the bread (1.5 cm) using a bread knife; one slice was used as a bread sample on day 0, and the firmness was measured using a probe at the center of the sliced bread. The remaining two slices, heat-sealed in aluminum foil bags, were stored in a refrigerator at 4 °C and evaluated for samples on day 1 and day 4, respectively. The conditions for measuring firmness were as follows: mode, measure force in compression; pre-test speed, 2.0 mm/s; test speed, 2.0 mm/s; post-test speed, 5.0 mm/s; probe, TA-AACC36 probe; and penetration distance, 15 mm.

### 2.7. Analyzing Quality Characteristics of Purple-Colored WWB with the Addition of Dough Improvers

Vital wheat gluten, vitamin C (ascorbic acid), two emulsifiers (DATEM and SSL), and two enzymes (amylase and xylanase) were used to investigate the effects of dough improvers on WWB quality. The tested dough improvers and concentrations in producing the bread are shown in Table 2. The bread-making procedure was the same as that described in Section 2.5. The quality characteristics of WWB with improvers were measured as described in Section 2.6.

### 2.8. Statistical Analysis

All experimental results were obtained from two or more repeated measurements, and statistical analysis was performed. Differences between samples were analyzed with one-way analysis of variance (ANOVA) using the SPSS Statistics (ver. 22.0, IBM, Corp., Armork, NY, USA) software. Significance was verified using Tukey’s multiple comparison test at *p* < 0.05.

## 3. Results and Discussion

### 3.1. Particle Size Distribution of Purple-Colored WWF

Figure 1 illustrates the particle size distribution in the purple-colored WWF. Group S exhibited significantly smaller particles compared to group L. Within each group, the particle size of the WWF decreased as the rotor speed increased (group S: SH < SM < SL; group L: LH < LM < LL). The average diameters (d50) of particles representing 50% of the total volume were 115 μm, 185 μm, and 258 μm for SH, SM, and SL, respectively, while LH, LM, and LL had average diameters of 294 μm, 348 μm, and 492 μm, respectively. Group S also exhibited a significantly brighter appearance with fewer speckles than group L. This observation aligns with previous findings by Moon et al. [33], who reported a negative correlation between brightness and particle size in WWF. Ahmed et al. [34] similarly found that the brightness of wheat flour increased with greater light scattering as the surface area increased and the particle size decreased.

In general, the particle size of WWF can vary depending on the operating conditions of the mill, even when using the same milling equipment and wheat kernels. Jet milling with increased air pressure and a reduced input speed [11], as well as longer grinding times in roll milling [20], have been reported to produce WWF with smaller particle sizes. Consistent with these findings, our study revealed that the varying rotor speeds in the ultra-centrifugal mill influenced the particle size, resulting in visible differences in appearance.

### 3.2. SRC of Purple-Colored WWF

The moisture content of the WWF ranged from 11.5% to 11.9% for group S and from 12.6% to 12.7% for group L, and this difference between the groups was significant (*p* < 0.05). The variance in sieve size used in the ultra-centrifugal mill led to a longer milling time and more moisture evaporation in group L compared to group S. The ash content of the WWF was 1.4% for both groups, indicating no significant difference between the groups, regardless of the rotor speed of the mill. Typically, wheat minerals are primarily found in the bran layer [35], which can vary depending on the extraction ratio of the refined wheat flour. However, since the extraction ratio of the WWF was 100%, the ash content of the WWF was not affected by the different milling conditions when the same milling equipment and wheat kernels were used. 

Regarding the flour quality characteristics, the solvent retention capacity (SRC) values in water and sodium carbonate solutions of the WWF, representing water absorption and damaged starch contributions, respectively, are depicted in Figure 2. 

The water SRC value of the WWF ranged from 87.6% to 95.2% for group S and from 75.7% to 82.9% for group L. The sodium carbonate SRC value of the flour ranged from 112.2% to 123.7% for group S and from 93.2% to 99.5% for group L. Group S exhibited significantly higher SRC values in both water and sodium carbonate solutions compared to group L (*p* < 0.05). Previous studies [10,11] have demonstrated that as the particle size decreases, the water SRC value or absorption and the sodium carbonate SRC value of WWF milled from the same wheat kernels increase, which aligns with the findings of this study. However, within each group, the water and sodium carbonate SRC values of WWF increased as the particle size increased and with a decreasing rotor speed. This observation can be attributed to the reduced rotor speed in the ultra-centrifugal mill, coupled with the same sieve, resulting in a longer grinding time for the whole wheat kernels and consequently the increased production of damaged starch during the milling process. Although similar particle sizes in WWF can be achieved using different milling equipment, conditions, wheat varieties, and kernels, the generation of damaged starch during milling can vary significantly. Hence, it may be challenging to predict the amount of damaged starch solely based on the WWF particle size. Damaged starch in flour has higher water absorption than undamaged starch [15], which contributes to an increase in water absorption. Excessively damaged starch could negatively affect bread quality by delaying the water absorption of gluten proteins and inhibiting gluten development [36].

### 3.3. SDS Sedimentation Volume in the Purple-Colored WWF

For the purple-colored WWF, the sodium dodecyl sulfate (SDS) sedimentation volume, which indicates gluten strength, is presented in Figure 3. Group S showed sedimentation volumes of 35.5–46.5 mL, 33.5–40.5 mL, and 31.5–38.5 mL at settling times of 20, 40, and 60 min, respectively. Group L exhibited sedimentation volumes of 26.5–35.5 mL, 24.0–32.5 mL, and 23.3–30.5 mL at the same settling times. The SDS sedimentation volume was significantly higher in group S than in group L, indicating stronger gluten strength. Within each group, a lower mill rotor speed corresponded to a lower SDS sedimentation volume, indicating weaker gluten strength [37]. However, the effect of the particle size on the gluten strength differed between the two groups, with a more significant impact observed in smaller particle sizes. It is worth noting that the wet gluten content decreased with increasing particle size, as reported by Bressiani et al. [10].

### 3.4. The Dough-Mixing Property of Purple-Colored WWF

Mixograms of the dough characteristics of the WWF are shown in Figure 4. Group S displayed a wider mixing band at the beginning of dough mixing, gradually narrowing after the mixing peak. In contrast, group L did not exhibit a distinct peak in the mixing band, and the band rapidly narrowed with continuous mixing, indicating weaker gluten strength. Notably, LL in group L demonstrated a narrow mixing band after 4 min, indicating the weakest gluten strength. In contrast, the SH of group S showed a wide mixing band at the beginning of mixing and maintained a relatively wide band even after the mixing peak, indicating relatively high gluten strength. Khatkar et al. [38] found that wheat flour with high gluten strength maintained a wider band even after the mixing peak, while wheat flour with low gluten strength exhibited a narrower band after the peak. This suggests that both the quantity and quality of gluten proteins in the flour influence the mixing patterns. The observed mixing patterns in this study highlight the impact of the particle size on gluten network formation during mixing. Bressiani et al. [10] also investigated the effect of the particle size on the thermomechanical properties of WWF using a Mixolab and found a more pronounced effect of coarse particles on the gluten network, indicating a detrimental effect on bread-baking quality. Additionally, precise control of the water quantity and mixing time is crucial in producing high-quality bread. Based on the mixograph data in our study, the optimal water absorption levels for baking were determined as 7.5 g for group S and 7.0 g for group L. These water levels were influenced by the particle size and damaged starch content of the WWF.

### 3.5. Bread-Making Performance of Purple-Colored WWF

Cross-sectional photographs of the WWB are shown in Figure 5, and the quality characteristics of the WWB formulated using WWF are listed in Table 3. The dough height after fermentation ranged from 47.7 to 50.3 mm for group S and 40.3 to 48.6 mm for group L. The WWFs in group S had significantly higher dough heights than those in group L (*p* < 0.05) with respect to the rotor speed of the mill. The weight loss during baking was 8.3–8.7% for group S and 7.9–8.5% for group L, with no significant difference observed. The bread height ranged from 40.5 to 42.3 mm for group S and 32.6 to 41.4 mm for group L. The bread volume ranged from 307.8 to 326.8 mL for group S and 247.6 to 323.9 mL for group L. Similarly, the WWFs in group S exhibited significantly larger bread volumes than the corresponding flours in group L (*p* < 0.05). These results align with the trends predicted based on the particle size distribution and mean diameter of the WWF. The average particle diameter (d50) of WWF had a significantly negative correlation with the dough height, weight loss during baking, bread height, and bread volume (r = −0.953, −0.958, −0.922, and −0.907, respectively, *p* < 0.05). However, the SDS sedimentation volume at 20, 40, and 60 min had a significantly positive correlation with these quality parameters (r = 0.913–0.948, 0.950–0.963, 0.875–0.921, and 0.865–0.904, respectively, *p* < 0.05).

Regarding the bread crumb color, the *L**, *a**, and *b** values were 39.9–42.1, 14.3–14.9, and 20.2–22.0, respectively, for group S, and 43.9–44.4, 11.8–13.5, and 20.7–22.4, respectively, for group L. The WWFs in group S exhibited significantly lower *L** (lightness) and higher *a** (redness) than those in group L (*p* < 0.05). In each group, a higher dough height after fermentation and bread volume were observed when using the WWF prepared with an increased rotor speed, which is consistent with the findings reported by Khalid et al., regarding WWF produced using an ultra-centrifugal mill [39]. Therefore, damaged starch generated during milling in WWF can be considered a crucial factor affecting the bread volume [36]. Additionally, considering the influence of the particle size, the bread volume in group S was larger than that in group L, as evidenced by the SDS sedimentation volume, gluten strength, and development observed in the mixogram. In summary, damaged starch and the WWF particle size were identified as key factors influencing the quality characteristics of WWB. 

Changes in bread firmness during storage at 4 °C are shown in Figure 6. The group S WWB exhibited lower firmness compared to the group L WWB, indicating a negative correlation between bread firmness and bread volume. Moreover, the increase in firmness during storage was less pronounced in the group S WWB compared to the group L WWB, suggesting a lower tendency toward aging.

### 3.6. Quality Characteristics of Purple-Colored WWB with the Addition of Dough Improvers

The SH and LH WWF WWBs exhibited favorable quality characteristics, which were then used to investigate the effects of dough improvers on bread quality. Cross-sectional photographs and the quality characteristics of these WWBs with the addition of dough improvers are shown in Figure 7 and Table 4, Table 5 and Table 6. The influence of vital wheat gluten on bread appearance was evident in the cross-sectional photographs, whereas the effect of amylase was negligible.

As shown in Table 4, the dough height, bread height, and bread volume increased as the amount of vital wheat gluten increased. Previous studies by Indrani and Rao [14] and Boz and Karaogiu [15] reported a significant increase in WWB loaf volume with the addition of 2–2.5% vital wheat gluten. Boz et al. [16] also observed increases in dough water absorption, resistance to extension, extensibility, and adhesion when vital gluten was added to whole wheat dough. Although the addition of 10% vital wheat gluten was relatively high compared to common practice in the baking industry, the bread made with SH WWF (referred to as SH WWB) exhibited significant increases in bread height from 41.6 mm to 50.4 mm and in bread volume from 320.4 mL to 403.5 mL. In contrast, the bread made with the LH WWF (referred as LH WWB) showed a relatively insignificant increase in bread height from 41.4 mm to 42.9 mm and in bread volume from 323.9 mL to 346.7 mL. The results indicate that the effect of vital wheat gluten as a dough improver varies according to the particle size of the WWF, showing a positive effect in WWF with small bran particles and no effect in WWF with large bran particles. The presence of large bran particles can weaken the gluten network rather than facilitate gluten development, resulting in a diminished influence, even with the addition of vital wheat gluten.

The addition of vitamin C to WWB production resulted in a less significant quality improvement compared to the addition of vital wheat gluten (Table 4). Similarly, Indrani and Rao [14] reported a slight increase in WWB volume with the addition of 100 or 200 ppm ascorbic acid. However, the effect of vitamin C varied with the particle size of the WWB, as observed with vital wheat gluten. When 50 ppm of vitamin C was added, the SH WWB showed significant increases in height from 41.6 mm to 46.7 mm and in volume from 320.4 mL to 371.4 mL. In comparison, the LH WWB showed a relatively insignificant increase in height from 41.4 mm to 44.2 mm and in volume from 323.9 mL to 330.0 mL. The results suggested that the most appropriate level of vitamin C for quality improvement was 50 ppm. The oxidized form of ascorbic acid participates in oxidation reactions during the flour–water mixing process, such as the SH/SS exchange between the cysteine residues of the gluten-forming proteins. The oxidation of thiol groups promotes the formation of disulfide bonds between proteins, leading to gluten cross-linking and polymerization, which strengthens the gluten network [40].

Table 5 demonstrates that the addition of emulsifiers DATEM and SSL led to a slight increase in the height and volume of WWB. The trend observed in the improving effect of emulsifiers on bread formulated with WWF of different particle sizes was similar to that of vital wheat gluten and vitamin C. The SH WWB with emulsifiers showed a significant improvement compared to the LH WWB. The improvement in bread quality was more pronounced with the addition of SSL compared to DATEM. With the addition of DATEM (1%), the SH WWB increased in height from 41.6 mm to 43.3 mm and in volume from 320.4 mL to 340.4 mL, while the LH WWB showed no change or even decreases in height from 41.4 mm to 39.9 mm and in volume from 323.9 mL to 310.6 mm. With the addition of SSL (1%), the SH WWB increased in bread height from 41.6 mm to 46.7 mm and in bread volume from 320.4 mL to 357.5 mL, whereas the LH WWB only showed slight increases in height from 41.4 mm to 42.5 mm and in volume from 323.9 mL to 331.2 mm. Previous studies have reported the effects of DATEM on bread volume, where the loaf volume increased with increasing dough elasticity [19]. Additionally, when DATEM was combined with oxidizers, a synergistic effect was observed on the loaf volume of WWB [20]. DATEM and SSL have also demonstrated the greatest increase in loaf volume with the addition of 0.5% [16,41]. 

Table 6 reveals that the addition of xylanase and amylase did not significantly affect the height and volume of WWB. Only SH WWB showed a slight increase in height and volume. With the addition of 150 ppm xylanase, SH WWB increased in height from 41.6 mm to 42.2 mm and in volume from 320.4 mL to 348.0 mL. LH WWB decreased in height from 41.4 mm to 38.6 mm and in volume from 323.9 mL to 312.9 mL. Several studies have shown an increased loaf volume in WWB with the addition of xylanase [42,43,44,45,46], although our results did not show a significant effect. Xylanase is known to improve gluten hydration, network formation, and dough expansion during fermentation by reducing flour water absorption [43,44,46]. The particle sizes of whole wheat flour (158 and 261 μm) with the addition of 60 and 100 ppm xylanase content reduced the undesirable effects of fibers in the dough, leading to improved bread making [45]. An increased loaf volume could result from water transfer from pentoses to gluten, leading to the restructuring of the gluten network and increased dough expansion [46], less interference with gluten network formation by hydrolyzed low-molecular-weight hemicellulose [47], and the conversion of water-unextractable arabinoxylans to water-extractable arabinoxylans, leading to improved gas retention capacity [42]. However, the level and type of xylanase used also affect the WWB volume due to varied activities and action patterns, emphasizing the importance of optimizing the enzyme usage level [44]. On the other hand, with the addition of 15 ppm amylase, SH WWB increased in height from 41.6 mm to 41.7 mm and in volume from 320.4 mL to 348.9 mL. LH WWB decreased in height from 41.4 mm to 39.9 mm and volume from 323.9 mL to 328.1 mL. Armero and Collar [48], Bae et al. [49], and Boz et al. [16] reported an increased loaf volume in WWB after adding α-amylase. Amylase easily hydrolyzes damaged starch, generating free sugars that aid yeast fermentation. Since SH WWF contained more damaged starch than LH WWF, it was expected to result in a larger bread volume for SH WWB compared to LH WWB. In our study, the addition of xylanase and amylase demonstrated similar increases in bread volume and height, suggesting a comparable enhancement in WWB quality.

In summary, the study found that vital wheat gluten was the most effective dough improver in improving the quality of purple-colored WWB, followed by vitamin C, enzymes (amylase and xylanase), and emulsifiers (DATEM and SSL). The effect of dough improvers on bread quality was more pronounced when using WWF with a smaller particle size compared to WWF with a larger particle size. Therefore, controlling the particle size of purple-colored WWF and incorporating appropriate dough improvers can significantly enhance the quality of WWB. However, further optimization of these factors using response surface methodology is recommended to achieve even better bread quality. The study also highlighted the impact of the milling conditions on the WWF particle size and quality, with a 0.5 mm sieve opening and 14,000 rpm rotor speed being favorable for the production of WWF with superior bread-making performance. Overall, understanding and controlling these factors is crucial for the production of high-quality purple-colored WWB.

## 4. Conclusions

A purple-colored WWF was prepared by varying the milling conditions of an ultra-centrifugal mill. The sieve opening size and rotor speed during milling had a significant impact on the particle size and quality of the WWF. In terms of WWF quality, group S exhibited higher water absorption and a greater contribution of damaged starch, as determined by the SRC method, compared to group L. Additionally, the SDS sedimentation volume and dough mixing bandwidth of the WWF indicated that group S had better gluten-forming capabilities during mixing than group L. Regarding the performance of WWB in bread making, the volume and height of the WWB from group S were superior to those of group L, confirming the influence of the particle size on bread quality. Furthermore, the addition of dough improvers to WWB increased both the bread volume and height. Among the various dough improvers tested, vital wheat gluten proved to be more effective than vitamin C, enzymes (amylase and xylanase), and emulsifiers (DATEM and SSL). The enhancement in WWB quality through the addition of dough improvers was more pronounced in group S compared to group L. Based on the milling conditions investigated in this study, a sieve opening size of 0.5 mm and a rotor speed of 14,000 rpm were found to be favorable for the production of WWF with better bread-making performance. In conclusion, the control of the particle size of WWF and the adjustment of the formulation of WWB by incorporating dough improvers are critical factors in achieving excellent-quality purple-colored WWB.

## Figures and Tables

**Figure 1 foods-12-02591-f001:**
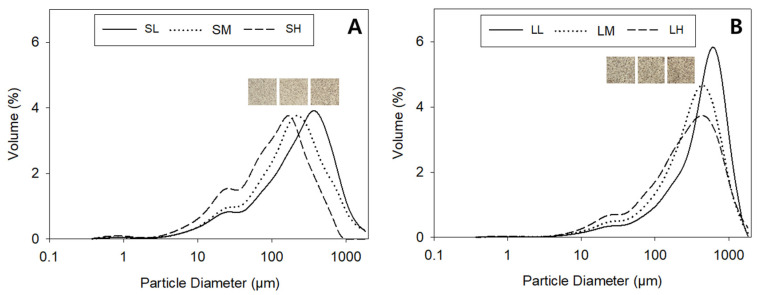
Particle size distribution in purple-colored WWF samples prepared at different milling conditions: (**A**) group S, SL, SM, and SH, 0.5 mm sieve opening and rotor speeds of 6000, 10,000, and 14,000 rpm, respectively; (**B**) group L, LL, LM, and LH, 1.0 mm sieve opening and rotor speeds of 6000, 10,000, and 14,000 rpm, respectively.

**Figure 2 foods-12-02591-f002:**
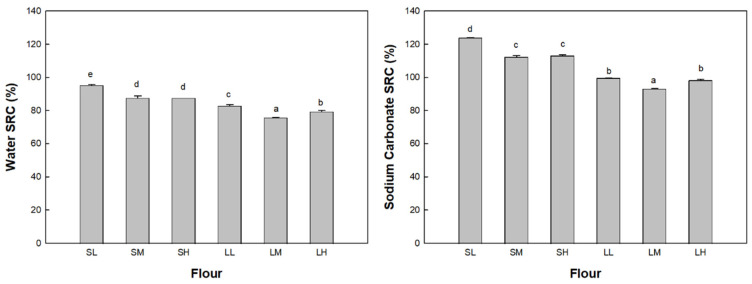
SRC of the flour samples: group S, SL, SM, and SH, 0.5 mm sieve opening and rotor speeds of 6000, 10,000, and 14,000 rpm, respectively; group L, LL, LM, and LH, 1.0 mm sieve opening and rotor speeds of 6000, 10,000, and 14,000 rpm, respectively. The same letters above the bars are not significantly different at *p* = 0.05, according to Tukey’s HSD test.

**Figure 3 foods-12-02591-f003:**
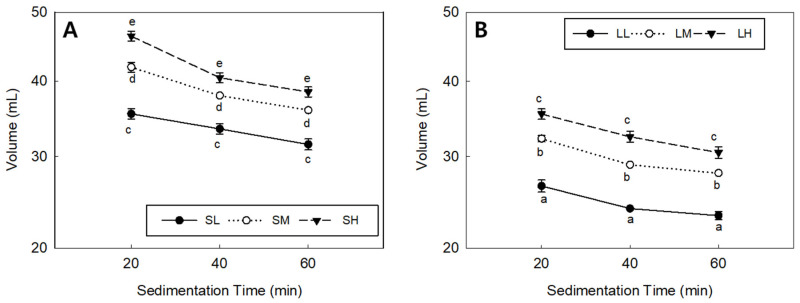
SDS sedimentation volume in the purple-colored WWF samples: (**A**) group S, SL, SM, and SH, 0.5 mm sieve opening and rotor speeds of 6000, 10,000, and 14,000 rpm, respectively; (**B**) group L, LL, LM, and LH, 1.0 mm sieve opening and rotor speeds of 6000, 10,000, and 14,000 rpm, respectively. The same letters above or below the symbols in line plots at the same sedimentation time are not significantly different at *p* = 0.05, according to Tukey’s HSD test.

**Figure 4 foods-12-02591-f004:**
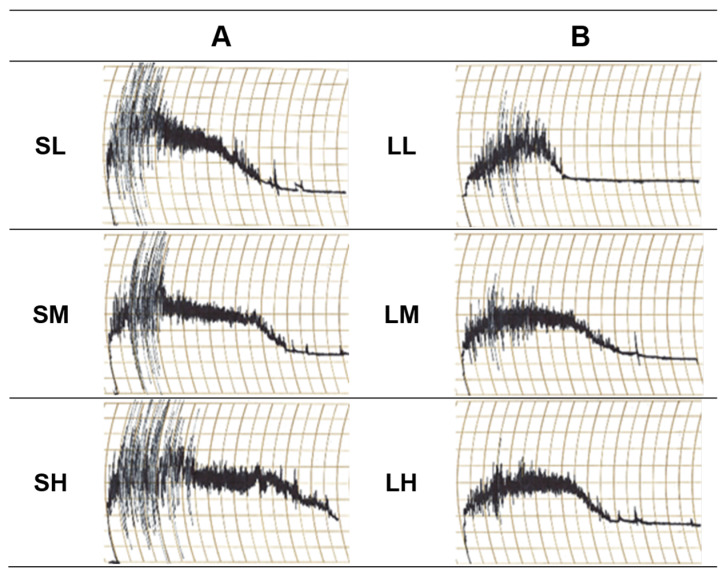
Mixograms of purple-colored WWF samples: (**A**) group S, SL, SM, and SH, 0.5 mm sieve opening and rotor speeds of 6000, 10,000, and 14,000 rpm, respectively; (**B**) group L, LL, LM, and LH, 1.0 mm sieve opening and rotor speeds of 6000, 10,000, and 14,000 rpm, respectively.

**Figure 5 foods-12-02591-f005:**
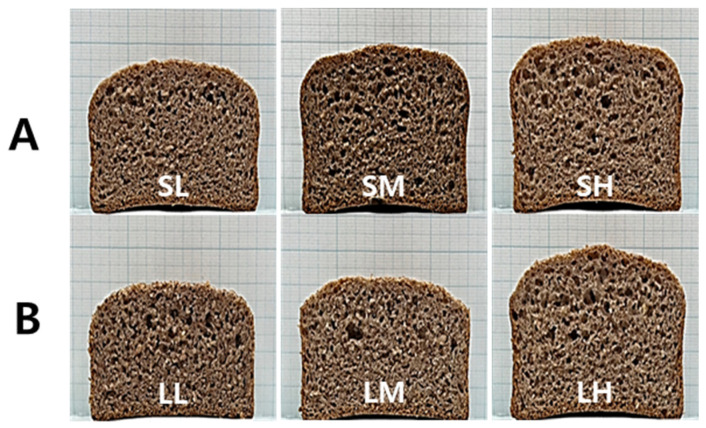
Side views and cross-sections of WWB made using purple-colored WWF samples: (**A**) group S, SL, SM, and SH, 0.5 mm sieve opening and rotor speeds of 6000, 10,000, and 14,000 rpm, respectively; (**B**) group L, LL, LM, and LH, 1.0 mm sieve opening and rotor speeds of 6000, 10,000, and 14,000 rpm, respectively.

**Figure 6 foods-12-02591-f006:**
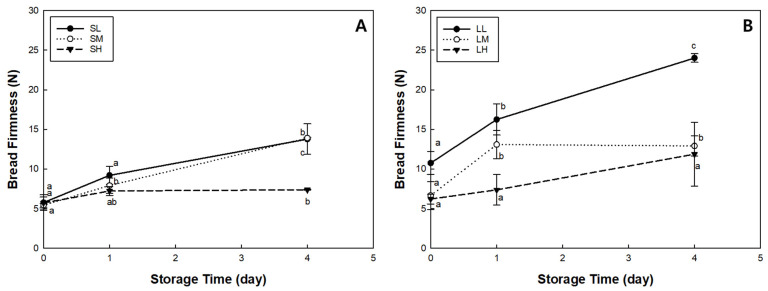
Changes in bread firmness during storage at 4 °C: (**A**) group S, SL, SM, and SH, 0.5 mm sieve opening and rotor speeds of 6000, 10,000, and 14,000 rpm, respectively; (**B**) group L, LL, LM, and LH, 1.0 mm sieve opening and rotor speeds of 6000, 10,000, and 14,000 rpm, respectively. The same letters near the symbols in the line plots are not significantly different at *p* = 0.05, according to Tukey’s HSD test.

**Figure 7 foods-12-02591-f007:**
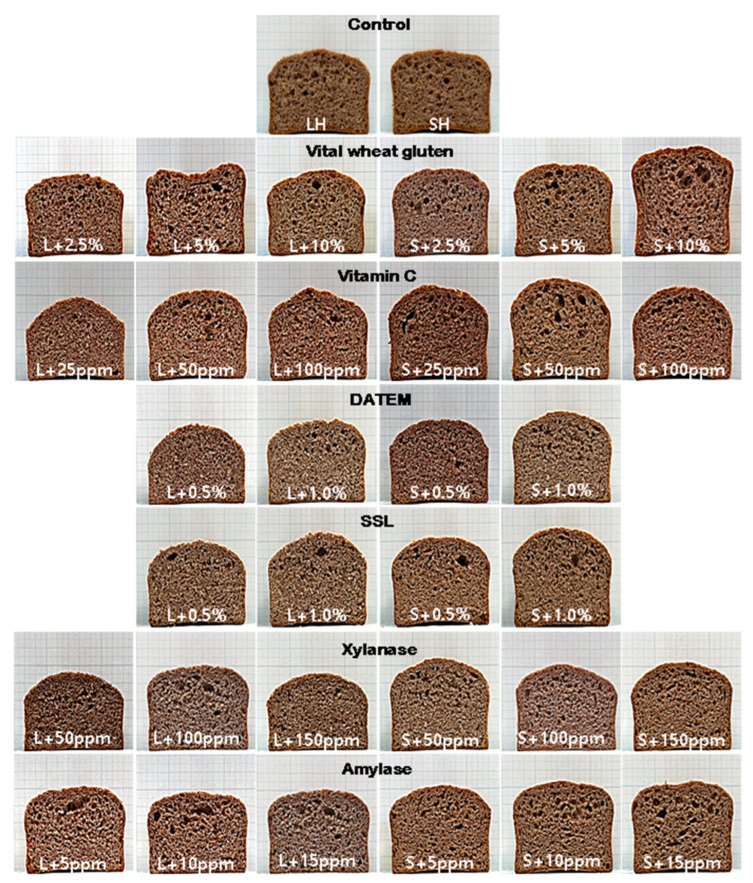
Cross-sections of purple-colored WWB with the addition of various dough improvers: LH, 1.0 mm sieve opening and a 14,000-rpm rotor speed; SH, 0.5 mm sieve opening and a 14,000-rpm rotor speed.

**Table 1 foods-12-02591-t001:** Formula and ingredients for bread prepared using AACC Method 10–10.03 with minor modification.

Ingredient	Group S	Group L
Flour (g)	100	100
Sucrose (g)	6	6
Non-fat dry milk (g)	4	4
NaCl (g)	1.5	1.5
Shortening (g)	3	3
Yeast (g)	2	2
Water (g)	75	70
Mixing time (s)	220	210
Fermentation time (min)	50	50

**Table 2 foods-12-02591-t002:** Concentrations of dough improvers used to produce WWB.

Improver		Tested Concentration
Vital wheat gluten (%)		2.5, 5.0, 10.0
Vitamin C (ppm)		25, 50, 100
Emulsifier (%)	DATEM	0.5, 1.0
	SSL	0.5, 1.0
Enzyme (ppm)	Amylase	5, 10, 15
	Xylanase	50, 100, 150

**Table 3 foods-12-02591-t003:** Quality characteristics of WWB formulated using purple-colored WWF samples and an ultra-centrifugal mill.

Sample		Dough Height (mm)	Weight Loss during Baking (%)	Bread Height (mm)	Bread Volume (mL)	*L**	*a**	*b**
Group S ^(1)^	SL	47.7 ± 1.3 ^c (2)^	8.3 ± 0.7 ^a^	40.5 ± 0.8 ^c^	307.8 ± 0.3 ^c^	41.3 ± 0.4 ^a^	14.9 ± 0.5 ^c^	22.0 ± 0.5 ^ab^
	SM	49.4 ± 1.2 ^cd^	8.6 ± 0.7 ^a^	41.6 ± 0.5 ^cd^	320.4 ± 2.3 ^d^	42.1 ± 1.4 ^ab^	14.3 ± 0.3 ^bc^	21.6 ± 0.4 ^ab^
	SH	50.3 ± 0.7 ^d (1)^	8.7 ± 0.1 ^a^	42.3 ± 0.3 ^d^	326.8 ± 0.8 ^e^	39.9 ± 1.4 ^a^	14.7 ± 0.1 ^c^	20.7 ± 0.7 ^a^
Group L	LL	40.3 ± 0.2 ^a^	7.9 ± 0.7 ^a^	32.6 ± 0.5 ^a^	247.6 ± 0.1 ^a^	44.4 ± 1.0 ^b^	11.8 ± 0.3 ^a^	20.7 ± 0.8 ^a^
	LM	44.6 ± 0.6 ^b^	8.3 ± 0.6 ^a^	36.2 ± 0.3 ^b^	281.1 ± 0.8 ^b^	43.9 ± 0.7 ^b^	13.2 ± 0.2 ^ab^	22.1 ± 0.6 ^ab^
	LH	48.6 ± 0.8 ^cd^	8.2 ± 0.5 ^a^	41.4 ± 0.5 ^cd^	323.9 ± 2.6 ^de^	44.0 ± 0.5 ^b^	13.5 ± 0.3 ^bc^	22.4 ± 0.1 ^b^

^(1)^ Group S, SL, SM, and SH, 0.5 mm sieve opening and rotor speeds of 6000, 10,000, and 14,000 rpm, respectively; Group L, LL, LM, and LH, 1.0 mm sieve opening and rotor speeds of 6000, 10,000, and 14,000 rpm, respectively. ^(2)^ Results are expressed as mean ± SD. Values with different superscript letters within the same column differed significantly (*p* < 0.05), according to Tukey’s HSD test.

**Table 4 foods-12-02591-t004:** Quality characteristics of purple-colored WWB with the addition of vital wheat gluten and vitamin C.

Sample	Vital Wheat Gluten				Vitamin C			
	Conc. (%)	Dough Height (mm)	Bread		Conc. (ppm)	Dough Height (mm)	Bread	
			Height (mm)	Volume (mL)			Height (mm)	Volume (mL)
SH ^(1)^	0	50.3 ± 0.7 ^b (2)^	41.6 ± 0.5 ^ab^	320.4 ± 2.3 ^a^	0	50.3 ± 0.7 ^ab^	41.6 ± 0.5 ^ab^	320.4 ± 2.3 ^ab^
	2.5	50.5 ± 1.3 ^b^	42.7 ± 0.5 ^b^	339.1 ± 3.6 ^ab^	25	54.7 ± 1.5 ^c^	48.1 ± 1.1 ^e^	367.2 ± 2.1 ^e^
	5.0	51.8 ± 2.4 ^bc^	45.6 ± 0.9 ^c^	365.2 ± 2.1 ^b^	50	52.7 ± 2.2 ^bc^	46.7 ± 1.9 ^de^	371.4 ± 0.3 ^e^
	10.0	54.1 ± 2.0 ^c^	50.4 ± 1.7 ^d^	403.5 ± 15.9 ^c^	100	51.4 ± 0.6 ^c^	45.6 ± 0.8 ^cd^	354.8 ± 2.4 ^d^
LH	0	48.6 ± 0.8 ^ab^	41.4 ± 0.5 ^ab^	323.9 ± 2.6 ^a^	0	48.6 ± 0.8 ^a^	41.4 ± 0.5 ^a^	323.9 ± 2.6 ^b^
	2.5	46.1 ± 2.2 ^a^	39.7 ± 0.9 ^a^	326.2 ± 2.2 ^a^	25	50.4 ± 1.4 ^ab^	40.4 ± 0.9 ^a^	314.8 ± 3.2 ^a^
	5.0	50.8 ± 1.4 ^bc^	40.2 ± 2.4 ^ab^	344.2 ± 18.0 ^ab^	50	51.3 ± 1.0 ^b^	44.2 ± 1.4 ^c^	333.0 ± 1.8 ^c^
	10.0	51.0 ± 1.3 ^bc^	42.9 ± 1.6 ^bc^	346.7 ± 17.4 ^ab^	100	50.9 ± 1.6 ^ab^	43.8 ± 1.2 ^bc^	338.0 ± 0.9 ^c^

^(1)^ SH, 0.5 mm sieve opening and a rotor speed of 14,000 rpm; LH, 1.0 mm sieve opening and a rotor speed of 14,000 rpm. ^(2)^ Results are expressed as mean ± SD. Values with different superscript letters within the same column differed significantly (*p* < 0.05), according to Tukey’s HSD test.

**Table 5 foods-12-02591-t005:** Quality characteristics of purple-colored WWB with the addition of emulsifiers.

Sample		DATEM			SSL		
	Conc. (%)	Dough Height (mm)	Bread		Dough Height (mm)	Bread	
			Height (mm)	Volume (mL)		Height (mm)	Volume (mL)
SH ^(1)^	0	50.3 ± 0.7 ^a (2)^	41.6 ± 0.5 ^bc^	320.4 ± 2.3 ^ab^	50.3 ± 0.7 ^b^	41.6 ± 0.5 ^a^	320.4 ± 2.3 ^a^
	0.5	45.3 ± 1.7 ^ab^	42.0 ± 1.0 ^c^	320.0 ± 10.0 ^ab^	45.9 ± 1.2 ^a^	43.3 ± 1.0 ^a^	338.3 ± 2.1 ^b^
	1.0	47.2 ± 1.2 ^a^	43.3 ± 1.2 ^c^	340.4 ± 3.4 ^b^	50.3 ± 1.2 ^b^	46.7 ± 1.3 ^b^	357.5 ± 4.0 ^c^
LH	0	48.6 ± 0.8 ^a^	41.4 ± 0.5 ^abc^	323.9 ± 2.6 ^ab^	48.6 ± 0.8 ^b^	41.4 ± 0.5 ^a^	323.9 ± 2.6 ^a^
	0.5	41.6 ± 5.6 ^a^	39.6 ± 0.8 ^a^	306.1 ± 19.5 ^a^	45.4 ± 1.6 ^a^	41.8 ± 1.3 ^a^	318.7 ± 7.1 ^a^
	1.0	45.2 ± 1.0 ^a^	39.9 ± 1.1 ^ab^	310.6 ± 8.8 ^a^	45.7 ± 0.9 ^a^	42.5 ± 1.4 ^a^	331.2 ± 6.7 ^ab^

^(1)^ SH, 0.5 mm sieve opening and a rotor speed of 14,000 rpm; LH, 1.0 mm sieve opening and a rotor speed of 14,000 rpm. ^(2)^ Results are expressed as mean ± SD. Values with different superscript letters within the same column differed significantly (*p* < 0.05), according to Tukey’s HSD test.

**Table 6 foods-12-02591-t006:** Quality characteristics of purple-colored WWB with the addition of enzymes.

Sample	Conc. (ppm)	Xylanase			Conc. (ppm)	Amylase		
		Dough Height (mm)	Bread			Dough Height (mm)	Bread	
			Height (mm)	Volume (mL)			Height (mm)	Volume (mL)
SH ^(1)^	0	50.3 ± 0.7 ^c (2)^	41.6 ± 0.5 ^bc^	320.4 ± 2.3 ^ab^	0	50.3 ± 0.7 ^d^	41.6 ± 0.5 ^abc^	320.4 ± 2.3 ^a^
	50	47.1 ± 3.4 ^abc^	42.6 ± 1.3 ^c^	354.4 ± 13.3 ^c^	5	49.0 ± 1.2 ^cd^	41.2 ± 1.4 ^abc^	338.9 ± 0.9 ^ab^
	100	42.1 ± 4.8 ^a^	39.0 ± 1.1 ^a^	322.6 ± 10.3 ^ab^	10	49.7 ± 1.3 ^cd^	42.2 ± 1.3 ^c^	349.4 ± 14.8 ^b^
	150	46.7 ± 3.9 ^abc^	42.2 ± 1.2 ^c^	348.0 ± 13.7 ^bc^	15	49.0 ± 1.8 ^cd^	41.7 ± 1.9 ^bc^	348.9 ± 17.0 ^b^
LH	0	48.6 ± 0.8 ^bc^	41.4 ± 0.5 ^bc^	323.9 ± 2.6 ^ab^	0	48.6 ± 0.8 ^bcd^	41.4 ± 0.5 ^abc^	323.9 ± 2.6 ^ab^
	50	42.4 ± 4.2 ^ab^	38.2 ± 0.7 ^a^	306.3 ± 4.8 ^a^	5	47.2 ± 1.2 ^abc^	39.6 ± 1.3 ^ab^	330.0 ± 8.2 ^ab^
	100	46.9 ± 1.6 ^abc^	40.1 ± 1.6 ^ab^	325.1 ± 8.2 ^ab^	10	45.9 ± 1.2 ^ab^	39.1 ± 1.5 ^a^	316.3 ± 5.8 ^a^
	150	42.0 ± 1.0 ^a^	38.6 ± 0.6 ^a^	312.9 ± 13.7 ^a^	15	44.7 ± 2.4 ^a^	39.9 ± 0.8 ^abc^	328.1 ± 1.6 ^ab^

^(1)^ SH, 0.5 mm sieve opening and a rotor speed of 14,000 rpm; LH, 1.0 mm sieve opening and a rotor speed of 14,000 rpm. ^(2)^ Results are expressed as mean ± SD. Values with different superscript letters within the same column differed significantly (*p* < 0.05), according to Tukey’s HSD test.

## Data Availability

The data presented in this study are available upon request from the corresponding author.

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
