# Peer review of "Combined Effects of Particle Size and Dough Improvers on Improving the Quality of Purple-Colored Whole Wheat Bread"

_foods, 2023, doi:10.3390/foods12132591_

Round 1
Reviewer 1 Report
This manuscript reports combined influences of wheat grounding conditions and application of dough improvers on whole wheat flour bread-making quality. Whole wheat bread consumption is desirable due to the beneficial effects on human health. However, the consumption of whole wheat bread is restricted due to the poor bread-making quality. Especially, authors investigated the wheat processing and formulations for purple colored wheat which is known to have high nutritional value. Therefore, the information reported in this manuscript is highly valuable as it might contribute to increasing whole wheat bread consumption. There is no critical point which requires revision. The minor revision suggestions are as follows:
Line 284-- Water absorption and mix time are usually considered important in flour breadmaking quality evaluation. Addition of brief discussion on variation of baking water absorption and mix time in relation to particle size and damaged starch content will improve content.
Although correlation coefficient is helpful to see associations between traits, no correlation was shown between flour quality traits and bread characteristics in this manuscript. The correlation coefficients which were highly significant better be commented in text to clarify the associations.
In conclusion, author better include the recommended wheat grinding condition and dough improver with justification.
Line 539-- 36. Joo, O. S.; Jung, Y. M. Effects of Attrition Milling in Wheat Flour on Starch Damaged of Dough and Bread Baking Properties. Korean J Postharvest Sci Technol, 2001, 8, 434-441. — Check if article title is correct.
Line 543--38. Khatkar, B. S.; Bell, A. E.; Schofield, J. D. A Comparative Study of the Inter-Relationships Between Mixograph Parameters and Bread- Making Qualities of Wheat Flours and Glutens. J Sci Food Agric 1996, 72, 71-85.
Author Response
Comments and Suggestions for Authors
This manuscript reports combined influences of wheat grounding conditions and application of dough improvers on whole wheat flour bread-making quality. Whole wheat bread consumption is desirable due to the beneficial effects on human health. However, the consumption of whole wheat bread is restricted due to the poor bread-making quality. Especially, authors investigated the wheat processing and formulations for purple colored wheat which is known to have high nutritional value. Therefore, the information reported in this manuscript is highly valuable as it might contribute to increasing whole wheat bread consumption. There is no critical point which requires revision. The minor revision suggestions are as follows:
Response) We appreciate the careful review and detailed comments the reviewer gave. We revised our manuscript marked with red color.
Line 284-- Water absorption and mix time are usually considered important in flour breadmaking quality evaluation. Addition of brief discussion on variation of baking water absorption and mix time in relation to particle size and damaged starch content will improve content.
Response) We explained the varied water absorption and mixing time-related to particle size and damaged starch.
Although correlation coefficient is helpful to see associations between traits, no correlation was shown between flour quality traits and bread characteristics in this manuscript. The correlation coefficients which were highly significant better be commented in text to clarify the associations.
Response) We added the correlation coefficients in interpreting the data.
In conclusion, author better include the recommended wheat grinding condition and dough improver with justification.
Response) We revised the conclusion as commented.
Line 539-- 36. Joo, O. S.; Jung, Y. M. Effects of Attrition Milling in Wheat Flour on Starch Damaged of Dough and Bread Baking Properties. Korean J Postharvest Sci Technol, 2001, 8, 434-441. — Check if article title is correct.
Response) We checked the article title, which is correct.
Line 543--38. Khatkar, B. S.; Bell, A. E.; Schofield, J. D. A Comparative Study of the Inter-Relationships Between Mixograph Parameters and Bread- Making Qualities of Wheat Flours and Glutens. J Sci Food Agric 1996, 72, 71-85.
Response) We corrected the article title.
Reviewer 2 Report
Comments for Authors
I carefully reviewed the current manuscript entitled “Combined effects of particle size and dough improvers for improving the quality of purple-colored whole wheat bread”. In this study, investigate the effect of particle size in purple-colored whole wheat flour, prepared using an ultracentrifuge mill under various conditions, on bread-making performance to produce more suitable whole wheat bread. The results declared that combination of controlling the particle size of purple-colored whole wheat flour and using dough improvers was effective in improving whole wheat bread quality. However, there are some weak points in the research, concerning overall design of the experiment and analytical chemistry. The objectives and significance of the study are not well described. what is the reason for choosing of purple-colored whole wheat flour in this study? I recommend the manuscript needs major revisions. My remarks about the text are as follows:
Abstract
The abstract should be concise, only discus the main theme of the article and current results and what conclusion you have reached from these findings.
Line 20-21, rewrite the statement “Improvement was significantly greater in group S than in 20 group L.” for clarity.
Introduction
This section must be reorganized. Some parts must be reduced so more focus can be given to the purple-colored whole wheat and especially nutritional perspectives of purple-colored whole wheat. Please mention previous studies regarding the quality of purple-colored whole wheat and their benefits or its contribution to bakery products.
Materials and Methods
As per objectives of current research line 79-83 “purple-colored WWF was assessed in terms of increased health benefits”. Where is nutritional analysis???? Did the authors evaluate the nutritional profile of purple-colored whole wheat?
Results and Discussion
Please improve the coherence in write-up and briefly describe that how this study helped to overcome these research gaps. In order to interpret the results, literature values must be compared. It is important to have a well-developed, thoughtful discussion.
Line 411, “leading to improved gas retention capacity.” Rephrase as “leading to Improve gas retention capacity.”
Conclusion
Line 437-440, The statement: “purple-colored wheat variety contains many functional components such as anthocyanin and polyphenol compounds found in wheat bran and has excellent antioxidant activity compared with common wheat” is not appropriate for a conclusion. The conclusion section must include your own findings.
Here is a summary of the results in conclusion. It is important to focus and target conclusions on the overall outcome of the study.
The manuscript must be improved in terms of English language, grammar, and syntax.
The manuscript must be improved in terms of English language, grammar, and syntax.
Author Response
Comments and Suggestions for Authors
Comments for Authors
I carefully reviewed the current manuscript entitled “Combined effects of particle size and dough improvers for improving the quality of purple-colored whole wheat bread”. In this study, investigate the effect of particle size in purple-colored whole wheat flour, prepared using an ultracentrifuge mill under various conditions, on bread-making performance to produce more suitable whole wheat bread. The results declared that combination of controlling the particle size of purple-colored whole wheat flour and using dough improvers was effective in improving whole wheat bread quality. However, there are some weak points in the research, concerning overall design of the experiment and analytical chemistry. The objectives and significance of the study are not well described. what is the reason for choosing of purple-colored whole wheat flour in this study? I recommend the manuscript needs major revisions. My remarks about the text are as follows:
Response) We appreciate the careful review and detailed comments the reviewer gave. We revised our manuscript marked with blue color.
Abstract
The abstract should be concise, only discus the main theme of the article and current results and what conclusion you have reached from these findings.
Ans) We revised the abstract as the reviewer commented.
Line 20-21, rewrite the statement “Improvement was significantly greater in group S than in 20 group L.” for clarity.
Response) We revised the statement.
Introduction
This section must be reorganized. Some parts must be reduced so more focus can be given to the purple-colored whole wheat and especially nutritional perspectives of purple-colored whole wheat. Please mention previous studies regarding the quality of purple-colored whole wheat and their benefits or its contribution to bakery products.
Response) We revised the introduction as the reviewer commented.
Materials and Methods
As per objectives of current research line 79-83 “purple-colored WWF was assessed in terms of increased health benefits”. Where is nutritional analysis???? Did the authors evaluate the nutritional profile of purple-colored whole wheat?
Response) We eliminated the statement and revised the objectives.
Results and Discussion
Please improve the coherence in write-up and briefly describe that how this study helped to overcome these research gaps. In order to interpret the results, literature values must be compared. It is important to have a well-developed, thoughtful discussion.
Response) We revised the results and discussion parts to improve the coherence in the write-up and added some limitations of the current study, which required further study.
Line 411, “leading to improved gas retention capacity.” Rephrase as “leading to Improve gas retention capacity.”
Response) We rephrased the part as commented.
Conclusion
Line 437-440, The statement: “purple-colored wheat variety contains many functional components such as anthocyanin and polyphenol compounds found in wheat bran and has excellent antioxidant activity compared with common wheat” is not appropriate for a conclusion. The conclusion section must include your own findings.
Response) We eliminated the statement and revised the conclusion.
Here is a summary of the results in conclusion. It is important to focus and target conclusions on the overall outcome of the study.
Response) We revised the conclusion.
The manuscript must be improved in terms of English language, grammar, and syntax.
Response) The manuscript was edited by a professional editor in Editage (a company provides editing services).
Comments on the Quality of English Language
The manuscript must be improved in terms of English language, grammar, and syntax.
Comments on the Quality of English Language
The manuscript must be improved in terms of English language, grammar, and syntax.
Response) The manuscript was edited by a professional editor in Editage (a company provides editing services).
Reviewer 3 Report
The manuscript evaluated the effects of combined control of whole wheat flour (WWF) particle size and the use of various dough improvers on the quality of purple whole wheat bread (WWB). I think the work is relevant, interesting, and done at a good methodological level. During the analysis of the manuscript, there were points that need to be corrected. Please find below a list of my recommendations:
1. Abstract must have rationale, objective, materials and methods, and conclusions. First sentence must be a rationale.
2. All acronyms must be spelled out in the Abstract. Such as S and L.
3. It is recommended that specific measured values be included in the Abstract.
4. Study needs to be hypothesis-driven.
5. The linkage between introduction paragraphs is missing.
6. There is a lack of adequate discussion of some of the findings, for example, in line 418: The quality improvement effects did not differ significantly between xylanase and amylase. Why? Please add the main reasons for the written sentence.
7. There is no space between the Celsius and the value, please check the full manuscript.
8. Given that the study presents a long list of abbreviations, I suggest adding a "glossary" table at the end of the paper or abbreviations part as it will aid the readers to learn about the concepts/terms that they are about to study.
9. Line 224: Replace “accords” with “accorded”.
10. L, a, and b should be in italics.
11. Please explain the meaning of the superscripts C2) and D1) in the footnote section of the table.
12. This Conclusion section is repetitive and should be rewritten. Please make sure your conclusions' section underscores the scientific value-added of your paper, and/or the applicability of your findings/results. Highlight the novelty of your study.
The manuscript evaluated the effects of combined control of whole wheat flour (WWF) particle size and the use of various dough improvers on the quality of purple whole wheat bread (WWB). I think the work is relevant, interesting, and done at a good methodological level. During the analysis of the manuscript, there were points that need to be corrected. Please find below a list of my recommendations:
1. Abstract must have rationale, objective, materials and methods, and conclusions. First sentence must be a rationale.
2. All acronyms must be spelled out in the Abstract. Such as S and L.
3. It is recommended that specific measured values be included in the Abstract.
4. Study needs to be hypothesis-driven.
5. The linkage between introduction paragraphs is missing.
6. There is a lack of adequate discussion of some of the findings, for example, in line 418: The quality improvement effects did not differ significantly between xylanase and amylase. Why? Please add the main reasons for the written sentence.
7. There is no space between the Celsius and the value, please check the full manuscript.
8. Given that the study presents a long list of abbreviations, I suggest adding a "glossary" table at the end of the paper or abbreviations part as it will aid the readers to learn about the concepts/terms that they are about to study.
9. Line 224: Replace “accords” with “accorded”.
10. L, a, and b should be in italics.
11. Please explain the meaning of the superscripts C2) and D1) in the footnote section of the table.
12. This Conclusion section is repetitive and should be rewritten. Please make sure your conclusions' section underscores the scientific value-added of your paper, and/or the applicability of your findings/results. Highlight the novelty of your study.
Author Response
Comments and Suggestions for Authors
The manuscript evaluated the effects of combined control of whole wheat flour (WWF) particle size and the use of various dough improvers on the quality of purple whole wheat bread (WWB). I think the work is relevant, interesting, and done at a good methodological level. During the analysis of the manuscript, there were points that need to be corrected. Please find below a list of my recommendations:
Response) We appreciate the careful review and detailed comments the reviewer gave. We revised our manuscript marked with purple color.
- Abstract must have rationale, objective, materials and methods, and conclusions. First sentence must be a rationale.
Response) We revise the abstract as the reviewer commented and add a rationale.
- All acronyms must be spelled out in the Abstract. Such as S and L.
Response) We spelled out all acronyms in the abstract.
- It is recommended that specific measured values be included in the Abstract.
Response) We added the measured values of particle size, which is essential.
- Study needs to be hypothesis-driven.
Response) We revised the introduction, mentioning the need for the study.
- The linkage between introduction paragraphs is missing.
Response) We revised the introduction for better linkage.
- There is a lack of adequate discussion of some of the findings, for example, in line 418: The quality improvement effects did not differ significantly between xylanase and amylase. Why? Please add the main reasons for the written sentence.
Response) We added the explanations.
- There is no space between the Celsius and the value, please check the full manuscript.
Response) We checked no space and corrected it.
- Given that the study presents a long list of abbreviations, I suggest adding a "glossary" table at the end of the paper or abbreviations part as it will aid the readers to learn about the concepts/terms that they are about to study.
Response) We added abbreviations at the end of paper.
- Line 224: Replace “accords” with “accorded”.
Response) We replaced “accords” with “accorded.”
- L, a, and b should be in italics.
Response) We revised L, a, and b to the italic font.
- Please explain the meaning of the superscripts C2) and D1) in the footnote section of the table.
Response) Superscripts a, b, c, d… are the statistical analysis results, which showed the difference between samples. 1), 2), 3).. are indicating the footnotes.
- This Conclusion section is repetitive and should be rewritten. Please make sure your conclusions' section underscores the scientific value-added of your paper, and/or the applicability of your findings/results. Highlight the novelty of your study.
Response) We revised the conclusion.
Reviewer 4 Report
The authors have done an interesting and detailed work on the " Combined effects of particle size and dough improvers for improving the quality of purple-colored whole wheat bread". The experimental design is novel and well executed. However, some problems in this paper need to be solved.
1. Why is the amount of water added different between the two experimental groups in Table 1? Whether to consider the influence of the difference in the amount of added water on the experimental content. What is the meaning of 0.1s difference in mixing time of WWF dough? Please explain it.
2. There are several formatting errors in the manuscript:
a. The first letter of the keywords should be capitalized,
b. Note the uniform font size in the legend and the chart. For example, in Figure 1, the legend "SM" and "LM" are larger than other identification sizes.
Author Response
Comments and Suggestions for Authors
The authors have done an interesting and detailed work on the " Combined effects of particle size and dough improvers for improving the quality of purple-colored whole wheat bread". The experimental design is novel and well executed. However, some problems in this paper need to be solved.
Response) We appreciate the careful review and detailed comments the reviewer gave. We revised our manuscript marked with green color.
- Why is the amount of water added different between the two experimental groups in Table 1? Whether to consider the influence of the difference in the amount of added water on the experimental content. What is the meaning of 0.1s difference in mixing time of WWF dough? Please explain it.
Response) In a bread baking study, water amount and mixing time are generally adjusted based on flour water absorption and gluten quality to achieve the best potential baking result. In our study, the water retention capacity of the flours in the two groups was different, and the dough mixing curve showed different mixing patterns, which were the bases for those adjustments.
- There are several formatting errors in the manuscript:
- The first letter of the keywords should be capitalized,
Response) We capitalized the first letter of the keywords.
- Note the uniform font size in the legend and the chart. For example, in Figure 1, the legend "SM" and "LM" are larger than other identification sizes.
Response) We revised the font size in the legend and the chart uniformly.
Round 2
Reviewer 3 Report
I have reviewed the revised manuscript and approved it.
Reviewer 4 Report
No further comment for revision.